# JNK regulates muscle remodeling via myostatin/SMAD inhibition

Sarah J. Lessard[1,2], Tara L. MacDonald[1,2], Prerana Pathak[1], Myoung Sook Han[3], Vernon G. Coffey [4,5], Johann Edge[6], Donato A. Rivas[7], Michael F. Hirshman[1], Roger J. Davis [3,8] & Laurie J. Goodyear[1,2]

Skeletal muscle has a remarkable plasticity to adapt and remodel in response to environmental cues, such as physical exercise. Endurance exercise stimulates improvements in muscle oxidative capacity, while resistance exercise induces muscle growth. Here we show that the c-Jun N-terminal kinase (JNK) is a molecular switch that when active, stimulates muscle fibers to grow, resulting in increased muscle mass. Conversely, when muscle JNK activation is suppressed, an alternative remodeling program is initiated, resulting in smaller, more oxidative muscle fibers, and enhanced aerobic fitness. When muscle is exposed to mechanical stress, JNK initiates muscle growth via phosphorylation of the transcription factor, SMAD2, at specific linker region residues leading to inhibition of the growth suppressor, myostatin. In human skeletal muscle, this JNK/SMAD signaling axis is activated by resistance exercise, but not endurance exercise. We conclude that JNK acts as a key mediator of muscle remodeling during exercise via regulation of myostatin/SMAD signaling.

[1] Research Division, Joslin Diabetes Center, Boston 02215 MA, USA. [2] Department of Medicine, Brigham and Women's Hospital and Harvard Medical School, Boston 02215 MA, USA. [3] Program in Molecular Medicine, University of Massachusetts Medical School, Worcester 01605 MA, USA. [4] Faculty of Health Sciences and Medicine, Bond University, Gold Coast 4226 QLD, Australia. [5] School of Medical Science, RMIT University, Melbourne 3000, Australia. [6] Massey University, Palmerston North 4442, New Zealand. [7] Jean Mayer USDA Human Nutrition Research Center on Aging at Tufts University, Boston 02111 MA, USA. [8] Howard Hughes Medical Institute, University of Massachusetts Medical School, Worcester 01605 MA, USA. Deceased: Johann Edge. Correspondence and requests for materials should be addressed to S.J.L. (email: sarah.lessard@joslin.harvard.edu)

Skeletal muscle is our largest organ by mass, and makes critical contributions to whole-body health due to its indispensable role in metabolism and mobility. Further contributing to the clinical importance of skeletal muscle is its remarkable plasticity to undergo specialized phenotypic adaptations, or remodeling events, in response to specific mechanical or molecular signals[1,2]. Remodeling of muscle toward an endurance phenotype, which includes increased capillary density and enhanced oxidative capacity, reduces the risk of metabolic diseases, such as type 2 diabetes and cardiovascular disease[3–5]. On the other hand, adaptations leading to muscle fiber hypertrophy and increased strength can offset morbidity and mortality associated with muscle wasting disorders that occur with age (sarcopenia) and cancer (cachexia)[6–8]. Endurance and hypertrophic adaptations represent two distinct classes of muscle remodeling programs that can lead to clinically significant changes in muscle phenotype and have profound effects on health and longevity. However, there is evidence that opposing signaling pathways may regulate muscle adaptation in response to endurance and resistance exercise[9–11]. Determining the precise mechanisms that contribute to these contrasting muscle remodeling programs is necessary to target muscle adaptation for optimal clinical benefit.

Muscle contraction and physical exercise are potent means to stimulate changes in muscle phenotype; thus contributing to the ability of exercise to act as an effective therapy for the prevention and treatment of numerous chronic diseases[4,12]. The molecular and morphological adaptations in skeletal muscle induced by exercise are specific to the modality of exercise undertaken[1,2]. Endurance, also known as aerobic, exercise (e.g., running, cycling) induces an "endurance" adaptive program in muscle, leading to increased cardiorespiratory fitness and metabolic health. In contrast, resistance exercise training (e.g., weight lifting) stimulates a hypertrophic adaptive program, leading to increased muscle mass and strength. Given the unparalleled ability of exercise to modify muscle phenotype, there is much interest in the development of pharmacological strategies for the treatment of chronic disease that mimic the effects of exercise. However, the mechanisms that allow for modality-specific muscle adaptations in response to endurance or resistance exercise are incompletely understood.

The adaptation of muscle to endurance or resistance exercise is a highly variable trait in humans and animals, with some individuals having greater adaptive responses to the same exercise stimulus than others[13–15]. Some of the heterogeneity in exercise response is believed to be due to inherited factors[13,16]. However, environmental factors such as advanced age and chronic disease can also impair the adaptive response of muscle to exercise, or physically limit the ability of an individual to undertake exercise training regimens[17]. Determining the molecular mechanisms that mediate muscle adaptation with exercise is a key strategy to improve muscle function in populations with low adaptive response or limited mobility. In addition, elucidating the signals that allow muscle to distinguish between endurance and hypertrophic adaptations will allow for the targeted development of therapies to induce the most appropriate adaptive program to treat a particular disease state or condition.

As a means to discover the molecular mechanisms that regulate endurance adaptations in skeletal muscle, our previous work utilized rodent models generated by selective breeding for low- or high-adaptive response to endurance exercise[16]. The failure to improve aerobic capacity in low responders to endurance training occurred in conjunction with a less oxidative muscle phenotype and deficiencies in exercise-induced angiogenesis in skeletal muscle[16]. Importantly, blunted endurance remodeling in the skeletal muscle of low responders to endurance exercise was associated with increased risk for chronic metabolic disease, including insulin resistance, dyslipidemia, and increased adiposity. Our data demonstrated that hyper-activation of the mitogen-activated protein kinase, c-Jun N-terminal kinase (JNK), was associated with the failure of muscle to undergo endurance remodeling with exercise. Thus, using these innovative genetic models of low and high adaptive response, we hypothesized that JNK activation during exercise is a negative regulator of endurance adaptations in muscle.

The present investigation aimed to directly test the hypothesis that JNK is a critical mediator of muscle remodeling. We employed a multi-disciplinary approach to determine the effect of JNK hyper-activation and loss of function on muscle phenotype and remodeling, including tissue culture systems, animal models, and human subjects. This work identifies JNK as a molecular switch that, when active, stimulates muscle fibers to grow, leading to increased muscle mass. Conversely, when JNK is inhibited, an alternative adaptive program is induced, leading to endurance adaptations and enhanced aerobic capacity. We find that JNK exerts its effects on muscle phenotype via phosphorylation of the transcription factor, SMAD2, at specific linker-region residues. JNK-mediated SMAD2 phosphorylation results in negative regulation of the myostatin/TGFβ pathway, thus allowing for muscle growth. In addition, we demonstrate that this novel signaling axis can be modulated by specific types of exercise in human skeletal muscle, therefore identifying JNK/SMAD signaling as a target to induce muscle remodeling. These data enhance our understanding of the fundamental mechanisms that mediate muscle reprogramming and remodeling in vivo.

## Results

**Loss-of-JNK enhances endurance remodeling in muscle.** Our previous work using models of high and low adaptive response to endurance exercise demonstrated that skeletal muscle JNK activation during exercise was negatively correlated to the adaptive response to endurance exercise[16]. Based on those data[16], we hypothesized that inhibition of JNK during exercise would enhance adaptive remodeling in response to endurance exercise in skeletal muscle. To test this hypothesis, an endurance exercise training study was performed in muscle-specific JNK1/2 knockout (mJNKKO; $Mapk8^{LoxP/LoxP}$ $Mapk9^{LoxP/LoxP}$ MCK-Cre$^{-/+}$) mice and wild-type (WT; MCK-Cre$^{-/+}$) controls. Mice were placed in cages with access to a running wheel, and running activity was electronically monitored for 10 week (Exercise-trained). A separate group of mice was kept in static cages for the experimental period (Sedentary). Total running distance was similar over the training duration in WT and mJNKKO mice (Supplementary Figure 1). Following the 10-week training period, endurance capacity was measured using a graded treadmill running test in all experimental groups as a physiological marker of endurance adaptation. Endurance capacity in trained mJNKKO was 45% higher compared to wild-type controls (Fig. 1a). Endurance capacity was similar in untrained WT and mJNKKO animals, demonstrating that increased endurance capacity in mJNKKO was due to enhanced adaptation to training, rather than intrinsic differences between genotypes (Fig. 1a). High endurance capacity, also known as cardiorespiratory fitness, is the gold-standard measure for the adaptive response to endurance training and has emerged as one of the best clinical markers for metabolic health and longevity in humans. These data demonstrate that skeletal muscle JNK activation during exercise is a negative regulator of improvements to endurance capacity.

Consistent with a phenotype of high-endurance capacity, mJNKKO had a higher capillary density (Fig. 1b, c) and a greater proportion of oxidative (type I) fibers in the red gastrocnemius muscle (Fig. 1d, e). In addition, exercise training caused a

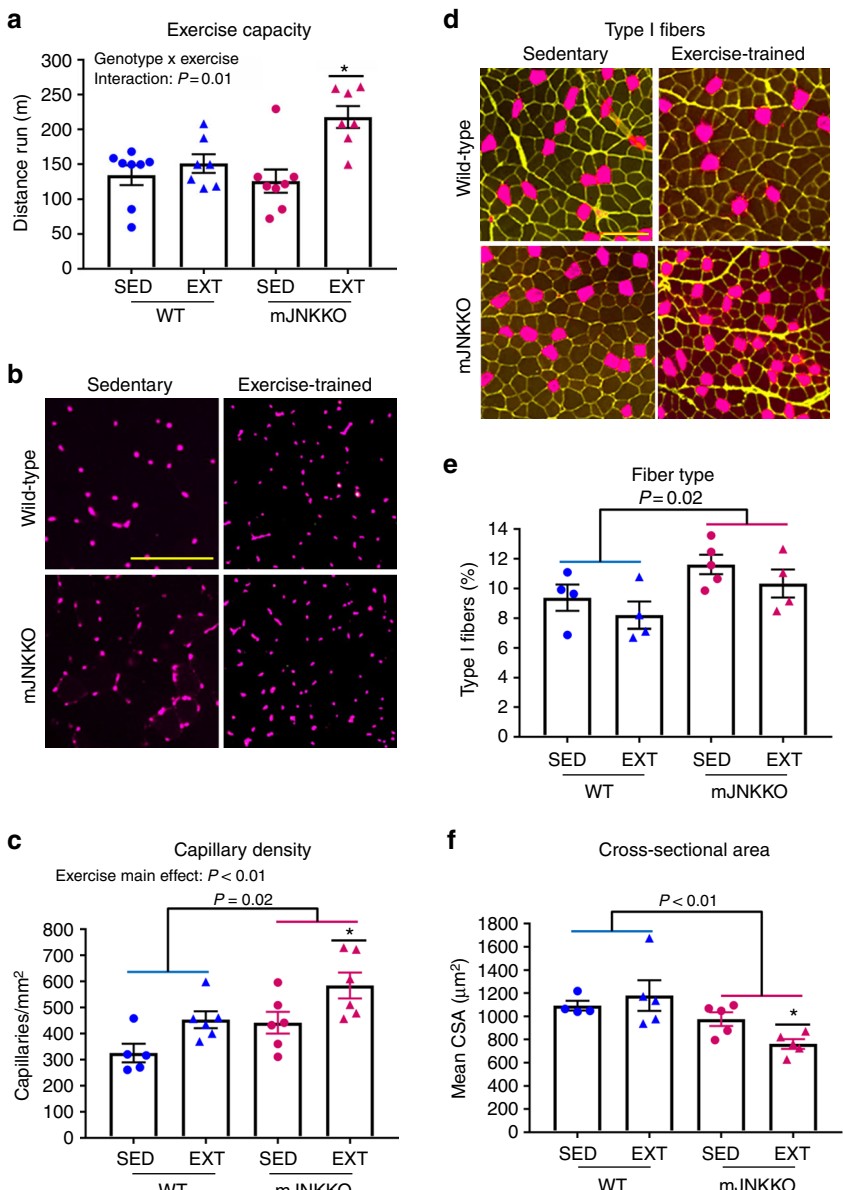

**Fig. 1** Loss of JNK in skeletal muscle enhances adaptation to endurance training. Muscle-specific JNK knockout mice (mJNKKO) and controls (WT; MCK-Cre$^{-/+}$) were placed in voluntary wheel-running cages for 10 weeks (Exercise-trained; EXT). A separate cohort of mice remained in cages without running wheels (Sedentary; SED) and acted as controls. **a** Exercise capacity was similar in WT and mJNKKO mice that were sedentary, but was higher in mJNKKO mice after training. **b**, **c** Histological analysis was performed on the red gastrocnemius muscle from sedentary and exercise-trained mJNKKO mice and controls. Staining with antibodies against CD31 (red) (**b**) was used to calculate capillary density (**c**). **d**–**f** Sections were stained with wheat germ agglutinin (WGA; green) and myosin heavy chain type I (red) antibodies (**f**) to determine fiber type (**e**) and cross-sectional area (**f**). *$P < 0.05$ vs SED of the same genotype by two-way ANOVA and Sidak's post hoc testing. Scale bars represent 100 μm. Each data point represents the results from an individual animal. Bar plots indicate mean ± SEM for all data

significant decrease in myofiber cross-sectional area in mJNKKO mice, but not in WT controls (Fig. 1f). Smaller myofiber cross-sectional area contributes to enhanced endurance capacity by allowing for better diffusion of oxygen and nutrients to working muscle[18]. In humans, small fiber cross-sectional area, higher capillary density, and increased oxidative myosin heavy chain expression characterize the muscle of elite endurance athletes[18]. Therefore, mJNKKO mice have a muscle phenotype that mimics that of endurance trained athletes. These experiments demonstrate that JNK is a negative regulator of endurance remodeling in muscle, and identify inhibition of JNK as a strategy to improve endurance adaptations.

**JNK is necessary for muscle hypertrophy**. It has been proposed that the signaling mechanisms regulating the remodeling programs for endurance or hypertrophic adaptations in muscle are antagonistic[9,10]. Indeed, concurrent training regimens employing both endurance and resistance exercise modalities can blunt muscle adaptation in humans[11]. Since JNK was a negative regulator of endurance training adaptations (Fig. 1), we hypothesized that JNK may be a positive regulator of hypertrophic adaptations in muscle. Functional overload (i.e., synergist ablation) is a surgical procedure that consistently induces muscle hypertrophy in mice, mimicking gains in muscle mass due to resistance training in humans. Functional overload of the plantaris muscle was

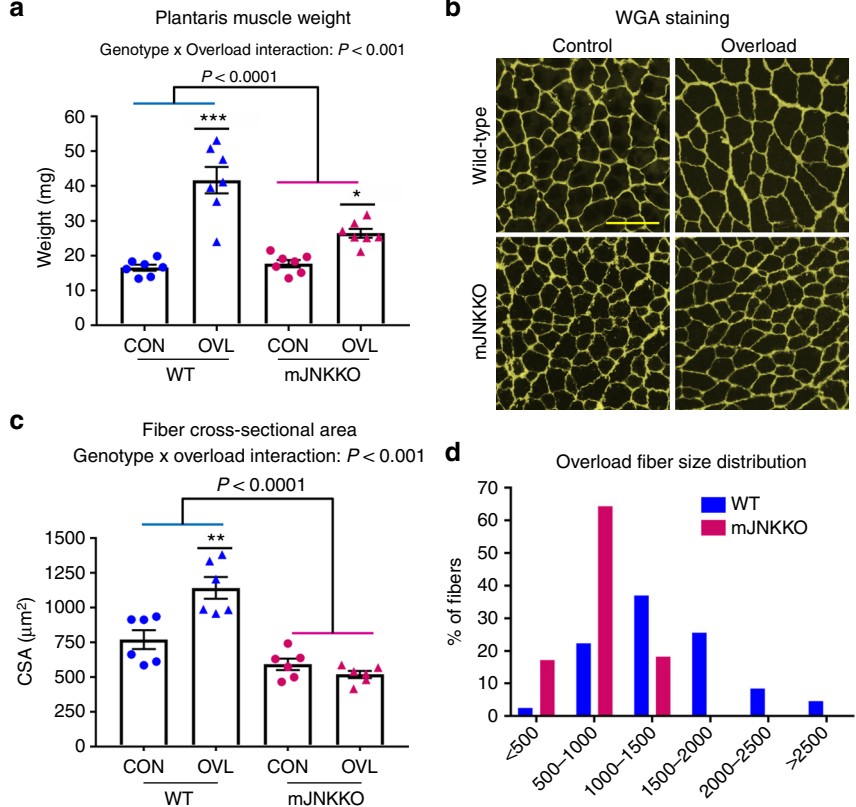

**Fig. 2** JNK is necessary for muscle hypertrophy with overload. The distal portion of the gastrocnemius muscle was surgically removed from one limb in muscle JNK knockout (mJNKKO) mice and wild-type controls (WT; MCK-Cre$^{-/+}$) to induce functional overload (OVL) of the plantaris muscle. The contralateral limb served as control (CON). **a** Fourteen days following overload surgery, the plantaris muscles were removed and wet weight was recorded. **b** Sections from plantaris muscles were stained with wheat germ agglutinin (green) for the calculation of fiber cross-sectional area (CSA). Scale bar represents 100 μm. **c**, **d** Average CSA was calculated (**c**) from the Control and Overload muscles from both genotypes, and fiber size distribution was plotted (**d**). *$P < 0.05$, **$P < 0.01$ vs Control from the same genotype by two-way ANOVA and Sidak's post hoc testing. Main effects of genotype are displayed with $P$-values. Each data point represents the results from an individual animal. Bar plots indicate mean ± SEM for all data

induced by partial removal of the gastrocnemius muscle in mJNKKO mice and WT controls to determine the effect of JNK knockout on muscle hypertrophy. Overload surgery was performed on one leg from each animal, and the contralateral leg served as a control. Two weeks of overload increased plantaris muscle weight by 2.5-fold in WT mice (Fig. 2a). However, the overload-induced increase in muscle weight was significantly blunted in mJNKKO mice (Fig. 2a). In addition, muscle fiber cross-sectional area was increased by 50% in WT animals, but JNK knockout prevented overload-induced increases in muscle cross-sectional area (Fig. 2b, c). These data demonstrate that JNK is necessary for increased muscle mass and myofiber hypertrophy in response to increased load, supporting the hypothesis that JNK is a positive regulator of hypertrophic adaptation in muscle.

**Muscle JNK activation coincides with SMAD2 phosphorylation.** Our data demonstrate that JNK acts as a positive regulator of hypertrophic adaptations and a negative regulator of endurance adaptations in muscle (Figs. 1 and 2). We next aimed to determine the mechanism by which JNK mediates muscle adaptive remodeling. Our previous work identified a novel exercise-activated phosphorylation site on the transcription factor, SMAD2, at a predicted JNK consensus sequence in its linker region (Ser245/250/255)[16]. As SMAD2 is a known regulator of muscle remodeling[19,20], we hypothesized that phosphorylation of SMAD2 may be a mechanism for JNK-mediated muscle remodeling with exercise. To determine whether JNK activation tracks with

SMAD2-linker (SMAD2-L) phosphorylation with exercise, we performed treadmill running and in situ contraction experiments in ICR (outbred) mice. SMAD2-L phosphorylation tracked closely with activation of JNK with both treadmill running and in situ contraction (Fig. 3a, b). However, treadmill running activated JNK/SMAD2-L phosphorylation to a relatively small extent (Fig. 3a) compared to in situ contraction, which more consistently and robustly induced JNK/SMAD2-L phosphorylation (Fig. 3b). While treadmill running induces metabolic stress in muscle via the utilization of energy substrates and ATP, in situ contraction elicits strong, eccentric contractions that induce both metabolic and mechanical stress. To determine whether mechanical stress alone is able to activate JNK/SMAD2-L signaling, we performed muscle stretch experiments. Passive stretch activated JNK and SMAD2-L robustly, while the metabolic stress indicator, AMPK, was minimally activated (Fig. 3c). Thus, JNK/SMAD phosphorylation is preferentially activated by contractions that elicit mechanical stress, rather than those producing metabolic stress (i.e., endurance exercise). Although other mitogen-activated protein kinases, p38 and ERK, were activated by muscle contraction and stretch; only JNK was activated in a similar pattern to pSMAD2-L. Therefore, our results identify JNK as a putative upstream kinase for SMAD2-L phosphorylation in muscle.

**JNK phosphorylates the SMAD2-linker region in muscle.** Muscles from WT and mJNKKO mice following in situ (tibialis anterior; TA) or in vitro (extensor digitorum longus; EDL)

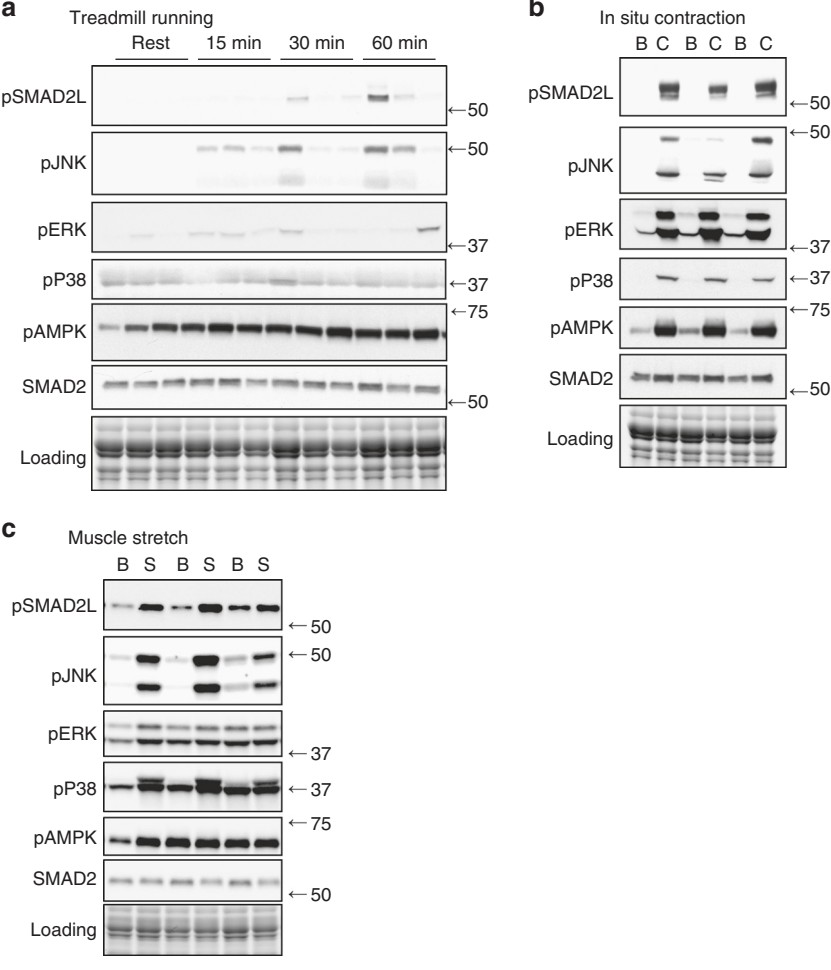

**Fig. 3** SMAD-linker and JNK phosphorylation with exercise and muscle contraction in mice. **a** ICR mice underwent moderate intensity treadmill running for 15, 30, or 60 min and gastrocnemius muscles were collected. Control (rest) mice did not undergo treadmill running. Western blotting was used to determine exercise-induced signal transduction. Data from $N = 3$ mice/group are shown. **b** Electrodes were used to stimulate the lower hindlimb muscles from ICR mice [C; contracted]. The contralateral limb was unstimulated and acted as a control [B; basal]. Data from $N = 3$ mice/group are shown. **c** Both soleus muscles were rapidly removed from mice and attached to a force tranducer in oxygenated Kreb's Henseleit Buffer. One muscle from each mouse was left at resting tension and acted as a basal control [Basal; B], while contralateral muscle was stretched for 10 min at a force of 0.12 N [Stretched; S]. $N = 6$ independent experiments were performed, and individual results from $N = 3$ are shown. pSMAD2L, linker region phosphorylated SMAD2; pJNK, phosphorylated (active) C-Jun N-terminal Kinase; pAMPK, phosphorylated (T172) AMP-activated protein kinase; pERK, phosphorylated extracellular signal regulated kinase; pP38, phosphorylated P38 Mitogen-Activated Protein Kinase; SMAD2, Total SMAD2. Images obtained using stain-free gel technology (Bio-Rad) that allows for total protein visualization and quantification are shown as a loading control (Loading)

contraction were analyzed to directly test the hypothesis that JNK is the upstream kinase for SMAD2-linker phosphorylation with exercise. In line with that hypothesis, SMAD2-L phosphorylation with muscle contraction was severely blunted in muscle-specific JNK1/2 knockout mice compared to WT controls (Fig. 4a–c). Next, C2C12 myoblasts were treated with the JNK activator Anisomycin (5 µM), to determine whether JNK activation is sufficient to induce SMAD2-L phosphorylation in muscle. Thirty minutes of Anisomycin treatment resulted in robust increases in JNK activation and SMAD2-linker phosphorylation (Fig. 4d). Skeletal muscle expresses two independent JNK isoforms, JNK1 and JNK2. Therefore, fusion plasmids for active JNK1 or JNK2 were expressed in C2C12 to determine whether JNK-associated SMAD2-L phosphorylation is isoform specific. Active JNK1 and JNK2 isoforms independently induced SMAD2-L phosphorylation in cultured muscle (Fig. 4e). SMAD2-linker phosphorylation was also increased following in vitro incubation of recombinant SMAD2 protein with purified immunoprecipitates of either active JNK1 or JNK2 (Fig. 4f). In contrast, we were unable to detect

JNK-mediated SMAD3 phosphorylation in its linker region at the Ser208 site using a commercially available antibody. In vitro phosphorylation indicates that JNK can directly interact with SMAD2 to phosphorylate its linker region in the absence of other cellular proteins. Taken together, these experiments (Fig. 4) demonstrate that JNK activation is both necessary and sufficient for SMAD2-L phosphorylation in muscle. Furthermore, we identify JNK as the primary upstream kinase for SMAD2-linker phosphorylation during exercise.

**JNK inhibits SMAD activity.** A JNK/SMAD signaling axis represents a novel exercise-activated pathway in skeletal muscle; introducing the possibility that JNK mediates hypertrophic vs. endurance remodeling via SMAD2 regulation. SMAD2 is a mediator of the transforming growth factor (TGF)-β family of ligands, which are critical regulators of remodeling in many tissues[21]. One of the most important TGFβ ligands for skeletal muscle remodeling is myostatin- a master regulator of muscle mass and metabolism in humans and animals that signals via

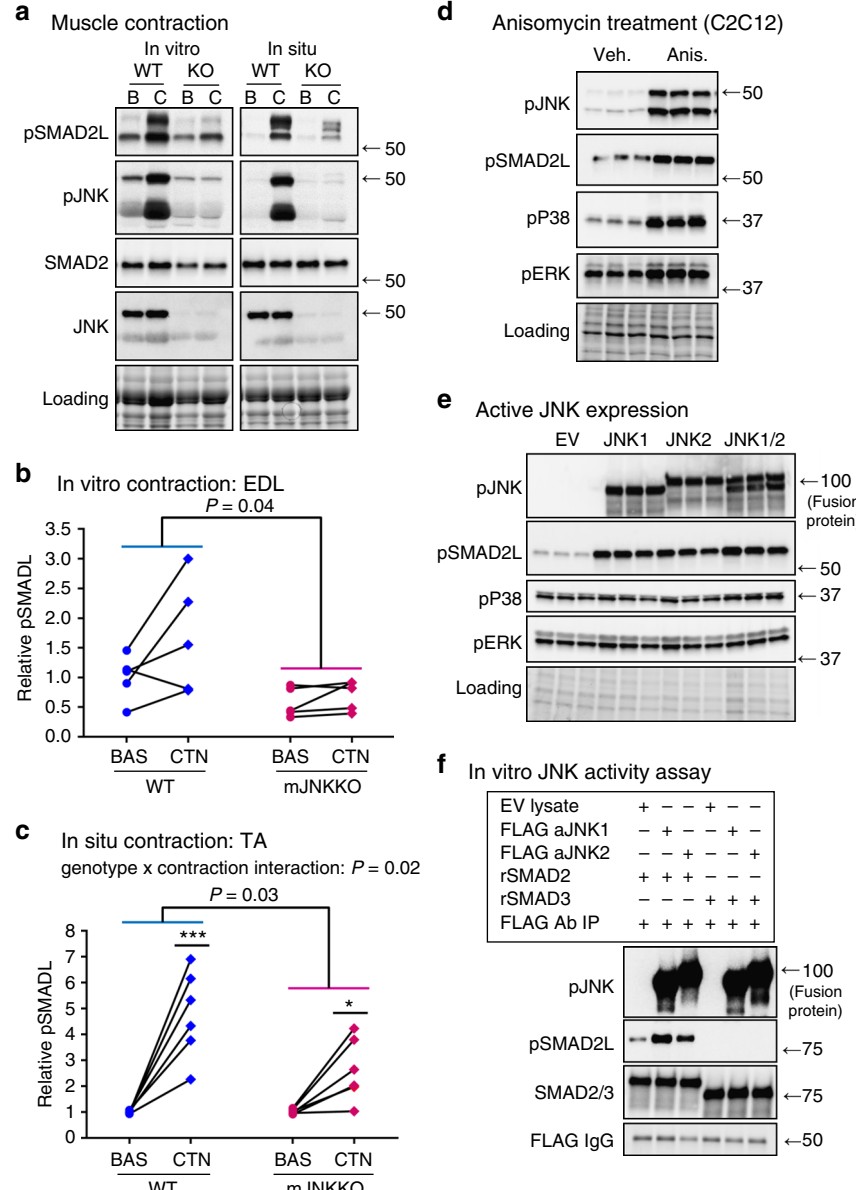

**Fig. 4** JNK is the upstream kinase for SMAD2 linker phosphorylation in muscle. **a–c** EDL and TA muscles from wild-type control (WT; MCK-Cre$^{-/+}$) and muscle-specific JNK1/2 knockout mice (KO) were stimulated via in vitro (EDL) and in situ (TA) contraction. Immunoblotting of phosphorylated and total SMAD2 and JNK was performed (**a**). SMAD2-linker (SMAD2L) phosphorylation in response to in vitro contraction (**b**) and in situ contraction (**c**) was blunted in muscle JNK knockout mice. Each data point represents the basal and contraction results from an individual animal joined by a line. *$P < 0.05$, ***$P < 0.01$ vs Basal from the same genotype by repeated measures two-way ANOVA and Sidak's post hoc testing. Main effects of genotype are displayed with $P$-values. **d** C2C12 myoblasts were treated with the JNK activator anisomycin (5 μM) for 30 min. **e** C2C12 myoblasts were transfected with plasmids expressing constitutively active JNK1, JNK2, or a combination of JNK1 and 2. Empty vector (EV) transfected cells were used as a control. **f** The ability of JNK to directly phosphorylate SMAD2 was assessed using an in vitro kinase assay. C2C12 lysates expressing empty vector (EV), or constitutively active JNK1 or JNK2 were purified by FLAG immunoprecipitation and incubated with recombinant SMAD2 and SMAD3 proteins. For tissue culture experiments (**d–f**), three independent experiments were performed and the data from one representative experiment, including all replicates is displayed. In all experiments, JNK activation (pJNK) and SMAD2 linker phosphorylation (pSMAD2L) were assessed by immunoblotting. Images obtained using stain-free gel technology (Bio-Rad) that allows for total protein visualization and quantification are shown as a loading control (Loading). pJNK phosphorylated (active) C-Jun N-terminal Kinase, pSMAD2L, linker-region phosphorylated SMAD2; SMAD2, total SMAD2; pP38, phosphorylation P38 MAPK; pERK, phosphorylated extracellular signal regulated kinase

SMAD2[22]. Therefore, we determined whether JNK-mediated SMAD2-linker phosphorylation regulates signaling or transcriptional activity in response to myostatin. To measure SMAD transcriptional activation, HEK293 cells were transfected with a SMAD binding element luciferase reporter (SBE4) in the presence or absence 20 nM myostatin. Myostatin induced a ~5-fold

increase in luciferase accumulation in control (Empty Vector) cells (Fig. 5a). However, cells expressing active JNK1 had severely blunted basal (Vehicle) and myostatin-mediated SMAD transcriptional activity, demonstrating that JNK activation has an inhibitory effect on SMAD activity. Impaired SMAD transcriptional activity was associated with increased SMAD2-L

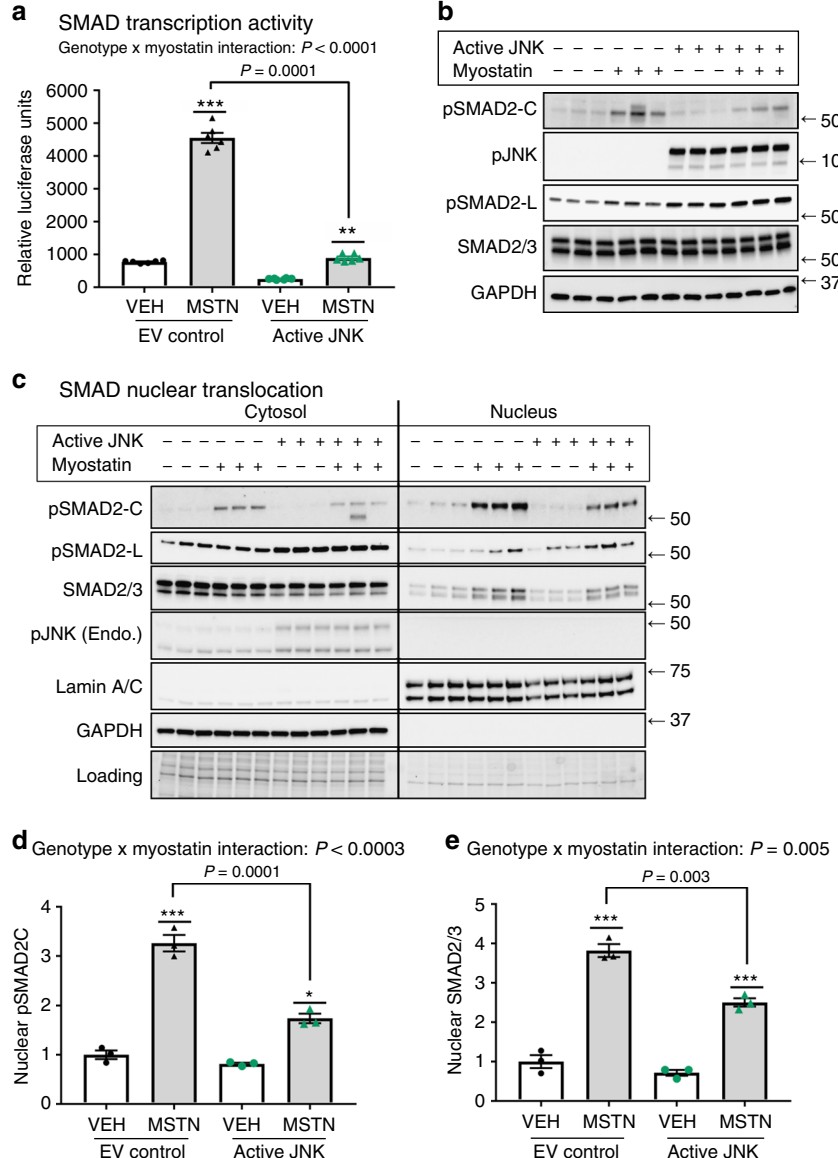

**Fig. 5** JNK inhibits myostatin/SMAD activity. **a** HEK293 cells were transfected with a SMAD-binding element (SBE4) luciferase reporter construct in the absence or presence of active JNK. 16 h following transfection, the cells were treated with 20 nM myostatin or Vehicle control for 4 h. Cells were lysed and analyzed for luciferase activity. **b** Western immunoblotting analysis was performed on whole-cell lysates from the luciferase experiment to assess levels of SMAD2 and JNK phosphorylation. **c** C2C12 myoblasts were transfected with empty vector (Control) or plasmids expressing active JNK fusion proteins. 24 h following transfection, the cells were treated with 20 nM myostatin or Vehicle for 30 min. Cytosolic and nuclear fractions were collected and immunoblotting was performed to assess the localization of total and phosphorylated SMAD2 and JNK. **d,e** Nuclear accumulation of phosphorylated SMAD2 (SMAD2C) (**d**) and total SMAD2 (**e**) were quantified. Data are representative of three independent experiments (**a**, **c**) and the data displayed are replicates from one representative experiment. pSMAD2-C, C-terminus phosphorylated SMAD2; pJNK, phosphorylated (active) c-Jun N-terminal Kinase; pSMAD2L, linker-region phosphorylated SMAD2; SMAD2/3, total SMAD2/3; GAPDH is a cytosolic loading control; Lamin A/C is a nuclear loading control. Stain-free gel images (Bio-Rad) are also shown as a loading control (Loading). *$P < 0.05$, **$P < 0.001$, ***$P < 0.0001$ vs vehicle treatment by two-way ANOVA with Sidak's post hoc testing. Main effects of genotype (JNK activity) are displayed. Bar plots indicate mean ± SEM for all data

phosphorylation in cells expressing active JNK (Fig. 5b). Thus, our data demonstrate that JNK activation initiates non-canonical phosphorylation of SMAD2 in its linker region, which has an inhibitory effect on SMAD2 activity.

**JNK inhibits myostatin-induced SMAD nuclear translocation**. In canonical SMAD signaling, TGFβ ligands (e.g., myostatin) bind to cell surface receptors, which initiate receptor-mediated phosphorylation of SMAD2 on its C-terminus (pSMAD2-C), followed by SMAD2 translocation to the nucleus and

transcriptional activation[23,24]. Active JNK reduced SMAD2-C phosphorylation in response to myostatin by ~20%, without altering SMAD2 protein levels (Fig. 5b). Therefore, impaired receptor-mediated phosphorylation may partly explain reduced SMAD transcriptional activity in the presence of active JNK. In some cellular contexts, linker-region phosphorylation of SMADs may prevent their translocation to the nucleus, and therefore transcriptional activity, even in the presence of receptor-mediated C-terminus phosphorylation[23]. Cytosolic and nuclear levels of SMAD2 were measured in C2C12 myoblasts to determine

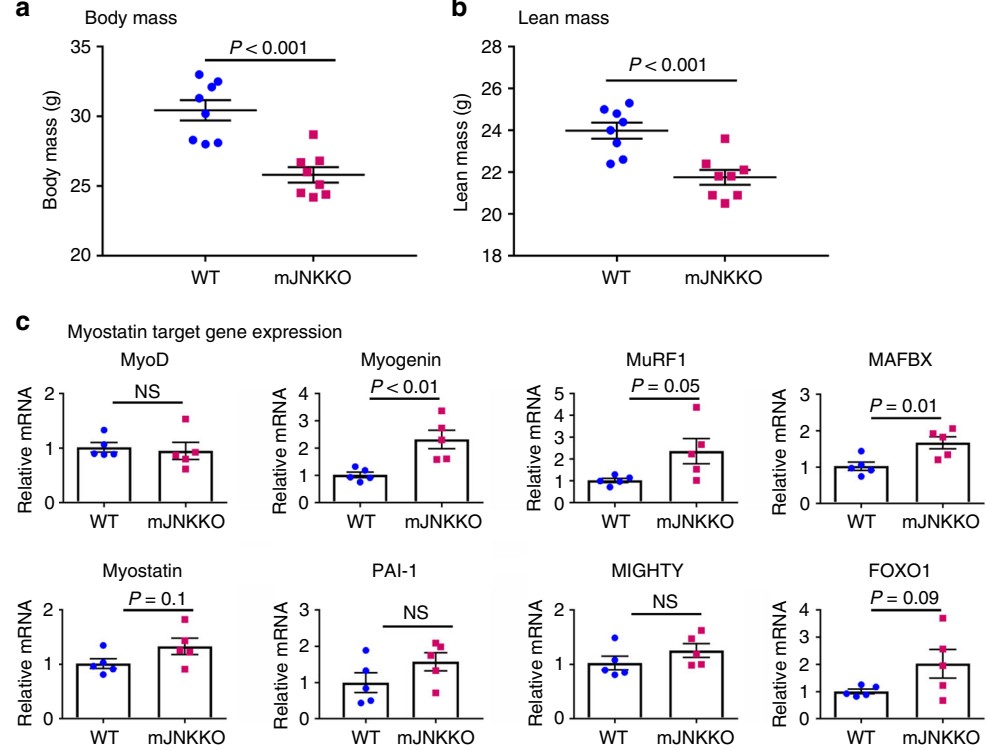

**Fig. 6** Lean mass and myogenic gene expression in mJNKKO mice. **a**, **b** Body weight was measured (**a**) and Lean mass (**b**) was calculated by DEXA in muscle-specific JNK1/2 knockout (mJNKKO) mice and wild-type (WT; MCK-Cre$^{-/+}$) controls. $N = 8$ per group. **c** mRNA was extracted from the gastrocnemius muscle of WT and mJNKKO mice and expression levels of myogenic regulators and myostatin target genes were measured using RT-PCR. mRNA expression is expressed relative to β2-microglobulin housekeeping gene. Each data point represents the results from an individual animal. Bar plots indicate mean ± SEM for all data. $P$-values derived from unpaired $t$-tests are shown for each measurement. NS, not significant ($P > 0.05$)

whether JNK-mediated SMAD linker phosphorylation prevents nuclear translocation in response to myostatin. Treatment with 20 nM myostatin for 30 min induced a threefold higher accumulation of pSMAD2-C and total SMAD2 in the nucleus of myoblasts (Fig. 5c–e). However, in cells overexpressing active JNK, the accumulation of SMAD2 in the nucleus was significantly blunted. Using an antibody that detects both SMAD2 and SMAD3 isoforms, we also demonstrate that SMAD3 accumulation in the nucleus was impaired in cells expressing active JNK. This indicates that reduced SMAD transcriptional activity with JNK overexpression (Fig. 5a), is likely due to effects on both the SMAD2 and SMAD3 isoforms. The effects of JNK activity on SMAD3 translocation may be due to direct phosphorylation at an unknown site, or through its complex formation with SMAD2[25]. Consistent with a role for SMAD linker phosphorylation in preventing nuclear translocation, the immunoblotting signal for pSMAD2-L was much higher in the cytosolic fraction of myocytes, while pSMAD2-C was enriched in the nuclear fraction (Fig. 5c). Thus, our results demonstrate that JNK-mediated phosphorylation of SMAD2 in its linker region inhibits transcriptional activity in response to myostatin by impairing both C-terminus phosphorylation and nuclear translocation of SMAD2/3 (Fig. 5).

**JNK suppresses myostatin activity in vivo**. Myostatin acts to suppress muscle growth, while myostatin inhibition via genetic or pharmacological means leads to muscle fiber hypertrophy in humans and animals[26]. Consistent with a role for JNK in myostatin inhibition, mJNKKO mice display lower body weight and lean mass compared to WT (Fig. 6a, b). Expression of myostatin mRNA in skeletal muscle was similar between WT and

mJNKKO mice (Fig. 6c). However, expression of the myostatin target genes, MuRF1 and MAFBX, were elevated in mJNKKO mice (Fig. 6c). Reduced lean mass and increased myostatin target gene expression in mJNKKO mice are consistent with evidence demonstrating JNK is a negative regulator of myostatin activity (Fig. 5).

**Resistance exercise activates JNK/SMAD signaling in humans**. Our data identify JNK activation as an inhibitor of TGFβ/myostatin signaling and a key mediator of muscle hypertrophy. To determine whether a JNK/SMAD2 signaling axis is activated in response to endurance or hypertrophic stimuli in humans, biopsies were taken from the vastus lateralis muscles of healthy young men at multiple time points before and after completion of a standard bout of acute endurance (cycling) or resistance (weighted leg extension) exercise. Resistance exercise robustly increased SMAD2-L phosphorylation ~4-fold in human skeletal muscle (Fig. 7a, b), with peak activation occurring immediately post-exercise. A secondary peak in SMAD2-L phosphorylation occurred 60 min post-exercise (Fig. 7b). JNK activation followed a very similar pattern to SMAD2-L phosphorylation with exercise (Fig. 7c). In contrast to resistance exercise, SMAD2-L and JNK were phosphorylated to a much smaller degree (<2-fold; not statistically significant) in response to endurance exercise in healthy humans (Fig. 7a–c). These data demonstrate that skeletal muscle JNK/SMAD2L signaling is upregulated by exercise in humans, and activation of this novel signaling axis is more consistent and robust following resistance exercise. Resistance exercise is the most effective modality for inducing muscle hypertrophy in humans. Thus, selective activation of JNK/SMAD signaling with resistance exercise is in line with data

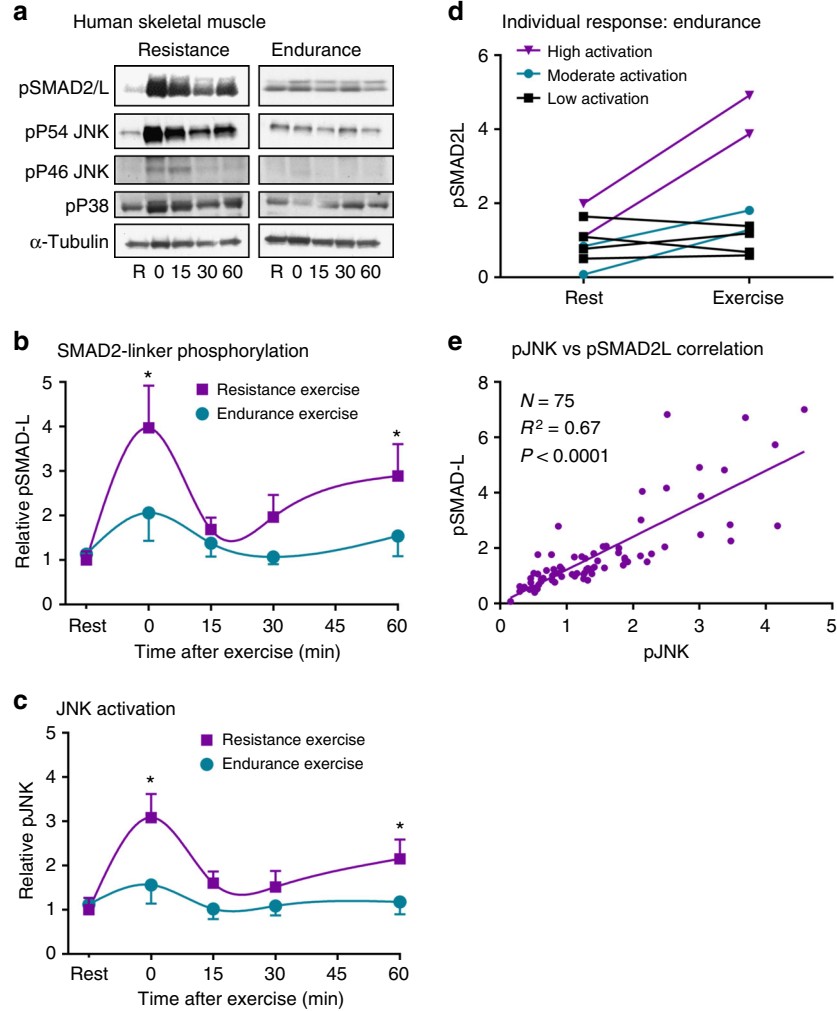

**Fig. 7** SMAD2-linker phosphorylation with exercise humans. **a** Healthy young men underwent a standard session of endurance exercise (cycling; $N = 8$ per time point) or resistance exercise (weighted leg extension; $N = 7$ per time point). Skeletal muscle biopsies were taken before exercise (R; rest), immediately post-exercise (Time 0), and 15, 30, and 60 min post-exercise. **b, c** The time course of SMAD2-linker phosphorylation (pSMAD-L) (**b**) and JNK activation (pJNK) (**c**) was determined. Mean ± SEM are shown for (**b**) and (**c**). **d** Individual levels of pSMAD2-L in resting (Rest) and immediately post-exercise (Exercise) biopsies for each subject that underwent endurance exercise are shown with values from the same subject joined by a line. **e** Linear regression analysis identified a significant correlation between JNK phosphorylation and pSMAD2-linker phosphorylation. *$P < 0.05$ vs Rest by two-way ANOVA and Tukey's Multiple Comparison Testing. pSMAD2L, linker-region phosphorylated SMAD2; pJNK, phosphorylated (active) C-Jun N-terminal Kinase. Different exposures are shown for the P54 and P46 splice forms of JNK, as the P54 band appeared darker in human skeletal muscle

demonstrating JNK is necessary for overload-induced muscle hypertrophy in mice (Fig. 2).

Although mean JNK/SMAD signaling was not significantly activated by endurance exercise in humans, the response was varied among subjects, with some individuals displaying no activation and others having >2-fold increases in phosphorylation (Fig. 7d). Heterogeneous activation of SMAD2-L phosphorylation with endurance exercise also occurs in animal models, and high-JNK/SMAD2-L activation predicts a poor adaptation to endurance training[16]. When data from all human muscle biopsies were pooled, patterns of SMAD2L phosphorylation mirrored JNK activation with exercise in humans, and linear regression analysis showed a significant correlation between pSMAD2-L and pJNK (Fig. 7e). Thus, we identify JNK-induced phosphorylation of SMAD2 in its linker region as a novel signaling axis in human skeletal muscle that is primarily activated by hypertrophic stimuli, such as resistance exercise. When considered collectively, our data support the hypothesis that activation of a JNK/SMAD2 signaling axis with resistance exercise induces muscle hypertrophy via TGFβ/myostatin inhibition; while suppression of JNK/SMAD signaling enhances endurance adaptations (Fig. 8).

## Discussion

Adaptive remodeling in skeletal muscle can produce clinically important changes in muscle morphology toward endurance or hypertrophic phenotypes. Elucidating the molecular signals that allow for increases in muscle fiber size or endurance has implications for the treatment of chronic diseases ranging from sarcopenia to diabetes. However, the precise mechanisms that determine these seemingly contrasting muscle adaptations are not fully understood. We have identified a novel role for the mitogen-activated protein kinase, JNK, as a key regulator of muscle adaptive remodeling in animals and humans. Furthermore, we provide mechanistic evidence that JNK modulates muscle phenotype via phosphorylation of SMAD2 and negative regulation of TGFβ/myostatin activity. This discovery represents a significant

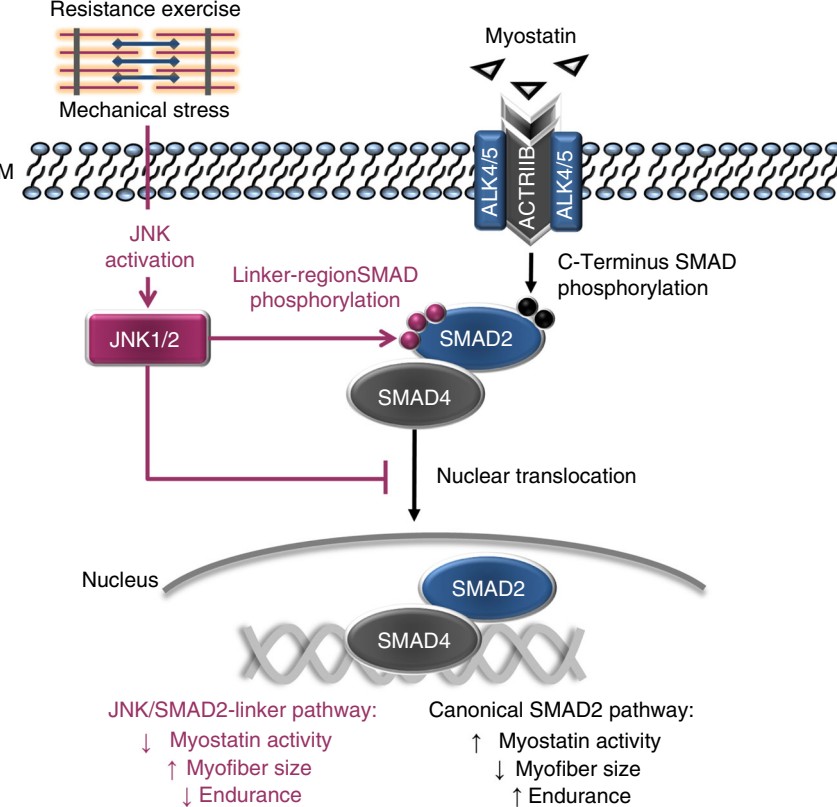

**Fig. 8** Hypothesized mechanisms by which JNK activation with exercise leads to muscle hypertrophy. Canonical myostatin signaling (depicted with black arrows) results in reduced myofiber size via receptor-mediated phosphorylation of SMAD2 on its C-terminus (Ser465/467). Myostatin-mediated SMAD2 phosphorylation induces dimerization with co-SMAD4 and translocation of the SMAD complex to the nucleus where DNA binding and transcription are initiated. Our data demonstrate that resistance exercise induces JNK activation and phosphorylation of SMAD2 in its linker region via a non-canonical pathway (depicted with red arrows). Phosphorylation of SMAD2 at specific linker region resides (Ser245/250/255) has an inhibitory effect on myostatin activity by preventing the nuclear translocation of SMAD2. Inhibition of myostatin via this novel JNK/SMAD pathway allows for exercise-induced muscle hypertrophy

advance in our understanding of the complex phenomenon of skeletal muscle remodeling.

To date, JNK has been known as an important regulator of cellular stress and inflammatory responses[27]. We now identify dual roles for JNK as a positive regulator of muscle hypertrophy and a negative regulator of endurance adaptation in muscle. Our observation that hypertrophy and endurance adaptations are regulated in opposing directions by JNK is in line with clinical observations that muscle adaptation is blunted when endurance and resistance training are performed concurrently (i.e., within the same exercise session)[11]. While there are some hypotheses regarding causes underlying interference between endurance and resistance adaptations[9,10], the precise molecular mechanisms are incompletely understood. Our data imply that JNK activation may be one mechanism for interference with concurrent training. When JNK was inappropriately activated with endurance exercise in models of low response to aerobic capacity, endurance adaptations were impaired[16]. Conversely, when JNK activation was prevented in mJNKKO mice, the mice had a potentiated response to endurance training, but muscle hypertrophy with overload was ablated (Figs. 1 and 2). Thus, JNK acts as a molecular switch to allow either endurance adaptations when inactive, or growth/hypertrophy adaptations when active. This premise may partly explain the mutual exclusivity of obtaining optimal endurance and strength adaptations.

While JNK is known to have several molecular targets, we identify SMAD2 as a novel exercise-activated JNK substrate. The discovery that JNK inhibits SMAD2 in muscle has important clinical implications, as SMAD2 is a downstream effector of myostatin- a TGFβ superfamily ligand[26]. Our data clearly show that JNK activation through multiple means, including pharmacological treatment, genetic manipulation, or muscle contraction leads to phosphorylation of the transcription factor SMAD2 at specific linker region residues. In turn, we provide evidence that JNK-mediated SMAD2 phosphorylation has an inhibitory effect on SMAD2 transcriptional activity and nuclear translocation. We also observed an inhibition of SMAD3 translocation to the nucleus in myocytes with JNK activation. The precise mechanisms for the inhibitory effect of JNK on SMAD3 have yet to be elucidated. The inhibitory effect of JNK on SMAD3 may be due to phosphorylation of SMAD3 by JNK on a yet undetermined residue(s), or and indirect effect via complex formation with SMAD2. However, it has previously been demonstrated that both SMAD2 and SMAD3 are necessary for the effects of TGFβ/myostatin on muscle mass[19,20,28]. Myostatin acts as a molecular brake that prevents muscle hypertrophy, while its inhibition allows for muscle growth[26,29]. Loss-of-function mutations in the myostatin gene, or inhibition of myostatin function via pharmacological antagonism, results in clinically significant increases in muscle mass in humans and animals[30,31]. Therefore, myostatin receptor antagonism is being explored as a treatment for muscle wasting diseases, such as cachexia and sarcopenia. Our work identifying JNK as a novel myostatin inhibitor provides a mechanism for our observation that mJNKKO mice have severely

impaired overload-induced hypertrophy (Fig. 2). Furthermore, we identify JNK activation as an alternative strategy for clinical myostatin inhibition.

In addition to playing a negative role in regulating muscle mass, studies in animal models suggest that myostatin may act as a positive regulator of endurance capacity[32]. In line with that assertion, myostatin inhibition causes muscle adaptations that would contribute to impaired endurance capacity, such as a shift toward a less oxidative muscle phenotype and lower capillary density[32,33]. These data suggest that myostatin may be necessary to maintain endurance adaptations in muscle, while myostatin inactivation can switch muscle toward the contrasting adaptive program of muscle growth. In support of a positive role for myostatin in endurance adaptations, we demonstrate that knockout of JNK in skeletal muscle, which would act to enhance myostatin activity, can increase endurance capacity and promote an endurance phenotype (i.e., oxidative fiber type and increased capillary density) in skeletal muscle. Furthermore, loss-of-muscle JNK activation with endurance exercise training in muscle JNK knockout mice led to a decrease in muscle fiber cross-sectional area, consistent with increased myostatin activity. Our work shows a novel functional intersection between two clinically important signaling networks: MAPK/JNK and TGFβ/myostatin.

Exercise induces thousands of molecular signals that contribute to its unparalleled ability to effectively treat and prevent a multitude of chronic diseases[4]. Accordingly, exercise-activated molecular targets such as AMPK and PGC1α have been extensively studied for their therapeutic potential as exercise mimetics[34]. While it has long been known that exercise can activate JNK in skeletal muscle[35,36], the role of JNK activation during exercise and its contribution to training-induced adaptations in muscle was not previously understood. We now demonstrate in humans, that a single bout of exercise can activate JNK leading to myostatin inhibition via JNK-mediated phosphorylation of SMAD2. In healthy humans, this novel JNK/SMAD2 signaling axis is primarily activated by resistance exercise, but not significantly activated by endurance (cycling) exercise (Fig. 7). Thus, exercise modalities that result in low levels of JNK activation (e.g., cycling; Fig. 7) would allow for myostatin to remain active and endurance adaptations to take place. Conversely, exercise that strongly activates JNK (e.g., weight lifting; Fig. 7) will result in inhibition of myostatin and a shift toward a hypertrophic adaptive program. Based on our data, we hypothesize that JNK/SMAD2 signaling acts as a mechanical sensor during exercise that determines whether the exercise modality necessitates hypertrophy-inducing adaptations, and therefore myostatin inhibition. In line with this hypothesis, mechanical stress (i.e., muscle stretch) was found to be a more potent activator of JNK than metabolic stress (i.e., endurance exercise) [Fig. 3]; and JNK has been demonstrated to have mechano-sensing properties in muscle and other tissues[37,38].

Although JNK/SMAD signaling is primarily activated by resistance exercise, a subset of individuals displays activation of JNK/SMAD signaling with endurance exercise (Figs. 3 & 7). In fact, we first discovered a potential JNK/SMAD signaling axis in samples taken following acute treadmill running exercise in our previous investigation of animal models of low and high adaptive response to endurance training[16]. Rat models selectively bred for an impaired adaptive response to endurance exercise had high activation of JNK/SMAD signaling with treadmill running compared to those selectively bred for high adaptive response to endurance exercise[16]. JNK/SMAD activation during acute exercise in low responders was associated with a failure to improve endurance exercise capacity with training, and blunted endurance adaptations in muscle with exercise. Thus, we propose that individuals with high JNK/SMAD activation in response to endurance exercise may fail to undergo endurance adaptations

due to inappropriate activation of a resistance exercise pathway. Indeed, we demonstrate that inhibition of JNK activation with endurance exercise in muscle JNK knockout mice greatly enhances endurance adaptations with training (Fig. 1).

The cause for increased JNK/SMAD signaling in some individuals during endurance exercise is unknown. However, heterogeneity in adaptation to endurance training and the phenomenon of "non-responders" to endurance exercise has been well-documented[13,15]. Our data suggest inappropriate JNK/SMAD activation as a putative mechanism for the failure to adapt to endurance training in a subgroup of the population. Low endurance exercise capacity in humans is associated with greatly increased risk for metabolic disease and mortality[39,40]. There is also extensive work linking JNK hyper-activation with chronic metabolic diseases such as obesity, insulin resistance, and diabetes, suggesting JNK plays a negative role in metabolic health[41,42]. In rodent models of low adaptive response to endurance exercise, activation of JNK signaling with endurance exercise was associated with increased risk factors for chronic metabolic disease, including glucose intolerance and increased adiposity[16]. Adaptation of muscle away from an oxidative/endurance phenotype in favor of a glycolytic/hypertrophic phenotype has been associated with increased risk for chronic metabolic disease in humans and animals[43,44]. Our data identify JNK hyper-activation as a potential mechanism for this shift in muscle phenotype associated with metabolic disease. Future studies should investigate whether causative links exist between metabolic health and JNK/SMAD signaling with endurance exercise, and whether inhibition of this pathway may act as a treatment for metabolic disease.

In summary, our data identify JNK/SMAD signaling as a molecular switch in skeletal muscle that induces muscle growth when active, and shifts muscle toward an endurance phenotype when inactive. Furthermore, we demonstrate regulation of TGFβ/myostatin signaling and activity as a mechanism for JNK-mediated effects on muscle phenotype, and validate the presence of this novel signaling axis in human skeletal muscle. This work elucidates the fundamental mechanisms that regulate the clinically important phenomenon of muscle remodeling and adaptation, and identifies JNK/SMAD signaling as a novel clinical target. As age and chronic disease can impair muscular adaptations to exercise, knowledge of the pathways that regulate adaptive reprogramming in muscle will be invaluable for designing strategies to improve muscle function in specific clinical populations.

## Methods

**Animal experiments**. Muscle-specific JNK1/2 knockout (mJNKKO) mice were generated on the C57BL/6J background using a Cre-Lox system under control of the muscle creatine kinase (MCK) promoter. $Mapk8^{LoxP/LoxP}$ mice[45] and $Mapk9^{LoxP/LoxP}$ mice[46] on a C57BL/6J strain background were used. C57BL/6J mice (stock# 000664) and B6.FVB (129S4)-Tg(Ckmm-cre)5Khn/J (stock# 006475)[47] were obtained from The Jackson Laboratory. mJNKKO mice were generated at the University of Massachusetts Medical School by M.S. Han and R.J. Davis, and all experiments were performed at the Joslin Diabetes Center. Genotype of mJNKKO and controls was determined by PCR and confirmed by the western blotting for total muscle JNK. Genomic DNA was genotyped by using a PCR-based procedure. Amplimers 5′-AGGATTTATGCCCTCTGCTTGTC-3′ and 5′-GAACCACTGTTCCAATTTC CATCC-3′ were used to detect $Jnk1^+$ (540 bp) and $Jnk1^{LoxP}$ (330 bp) alleles. Amplimers 5′-CCTCAGGAAGAAAGGGCTTATTTC-3′ and 5′-GAACCACTG TTCCAATTTCCATCC-3′ were used to detect $Jnk1^{LoxP}$ (1000 bp) and $Jnk1^{Δ}$ (410 bp) alleles. Amplimers 5′-GTTTTGTAAAGGGAGCCGAC-3′ and 5′-CCTGA CTACTGAGCCTGGTTTCTC-3′ were used to detect the $Mapk9^+$ (224 bp) and $Mapk9^{LoxP}$ alleles (264 bp). Amplimers 5′-GGAATGTTTGGTCCTTTAG-3′, 5′-GCTATTCAGAGTTAAGTG-3′, and 5′-TTCATTCTAAGCTCAGACTC-3′ were used to detect the $Mapk9^{LoxP}$ (560 bp) and $Mapk9^{Δ}$ alleles (400 bp). 5′-TTACT GACCGTACACCAAATTTGCCTGC-3′ and 5′-CCTGGCAGCGATGCTATT TTCCATGAGTG-3′ were used to detect $Cre\ recombinase$ (450 bp). All control mice (referred to as WT in the text) had the MCK-Cre$^{−/+}$ genotype and the mJNKKO mice had the Mapk8$^{LoxP/LoxP}$ Mapk9$^{LoxP/LoxP}$ MCK-Cre$^{−/+}$ genotype.

### Table 1 Mouse primer sequences used for analysis

| Gene | Forward | Reverse |
|---|---|---|
| MyoD1 | ACCAACGCTGATCGCCGCAA | GCAGCGGTCCAGGTGCGTAG |
| Myogenin | TGTGTCGGTGGACCGGAGGA | CCGCTGGTTGGGGTTGAGCA |
| MuRF1 | AAGCAGGTGCCACTCTCTGT | AGCTTCACACCTGTCCTTCG |
| MaFBx/Atrogin-1 | GACTGGACTTCTCGACTGCC | TCAGGGATGTGAGCTGTGAC |
| Myostatin | AGTGGATCTAAATGAGGGCAGT | GGAGTACCTCGTGTTTTGTCTC |
| Foxo1 | CCCAGGCCGGAGTTTAACC | GTTGCTCATAAAGTCGGTGCT |
| PAI-1 | ACAACCCGACAGAGACAATCC | TTCGTCCCAAATGAAGGCGT |
| Akirin1/Mighty | GTCTTCCAACTCCCGAGCAA | ACAGGCTTCGCTTTGACTGA |
| β2-microglobulin | CGGTCGCTTCAGTCGTCAGCATGG | CATTCTCCGGTGGGTGGCGTGAGT |

For treadmill running and in situ contraction experiments (Fig. 3), ICR mice were purchase from Charles River Laboratories. All mice were housed in a specific-pathogen-free facility, fed ad libitum with Purina Mouse Diet 9F 5020* (composition by Kcal%: 23.2% protein, 21.6% fat, 55.2% carbohydrate), and were maintained on a 12-h light/dark cycle. Body composition was measured in anesthetized mice using a Lunar PIXImus2 mouse densitometer. Male mice were used for all experiments. All animal experiments were approved by the Institutional Animal Care and Use Committee of the Joslin Diabetes Center.

**Endurance exercise training.** To determine the effect of muscle JNK on endurance training adaptations, WT and mJNKKO mice were housed individually in wheel-running cages and voluntary wheel-running distance was recorded daily for 10 weeks. Sedentary control mice were housed individually in static cages without running wheels. Endurance exercise capacity was measured in sedentary and exercise-trained mice during week 8 of the exercise training protocol using an incremental treadmill running test to exhaustion at a 15% grade. Endurance exercise capacity was expressed as total distance run during the exercise capacity test.

**Functional overload experiments.** As a model of resistance training, WT and mJNKKO mice ($N = 7$ per group) underwent unilateral ablation of the gastrocnemius muscle to induce functional overload and hypertrophy of the synergist plantaris muscle. Briefly, mice were anesthetized with pentobarbital and the fur from the distal hindlimb was removed. A 1 cm incision was made in the skin to isolate the gastrocnemius muscle tendon and the distal 1/3 of the gastrocnemius muscle was removed from one leg, leaving the plantaris and soleus muscles intact. The incision was closed with sutures and surgical glue and the mice were allowed to recover. The gastrocnemius muscle from the contralateral leg remained intact and acted as a control. Fourteen days following overload surgery, mice were killed and the plantaris muscles from the control and overloaded legs were removed, weighed, and frozen for histological analysis.

**Human subjects.** Sixteen, healthy young men (age $29.0 \pm 2.3$ yr) participated in this study and were randomly allocated to either 1) endurance exercise ($N = 8$) or 2) resistance exercise ($N = 8$) experimental groups. Detailed subject characteristics were previously published[48]. All subjects were recreationally active in various sports involving both endurance- and resistance-type exercise. Subjects assigned to endurance exercise undertook a preliminary incremental cycling test to volitional fatigue[49] to determine peak oxygen uptake (VO$_2$ peak, mean $\pm$ SD $54.3 \pm 3.6$ ml/kg/min). Resistance Exercise subjects performed a series of single-leg extension repetitions separated by 3 min recovery to establish individual one repetition maximum (1RM, mean $\pm$ SD $120 \pm 29$ kg). On a subsequent study visit, subjects were asked to abstain from vigorous physical activity for at least 48 h, and arrived at the laboratory following an overnight fast. Endurance Exercise subjects then underwent a standardized bout of exercise, consisting of 60 min cycling at ~70% of their previously determined VO$_2$ peak. Resistance Exercise subjects performed leg extension exercise for 8 sets of 5 repetitions at 80% of their individual 1RM, with each set separated by 3 min recovery. Skeletal muscle biopsies were taken from separate incisions (distal to proximal) of the vastus lateralis muscle from the same leg before the exercise bout (Rest), immediately post exercise (0 min), and 15, 30, and 60 min post exercise from each subject to determine exercise activation of JNK/SMAD signaling. Muscle samples were blotted and immediately frozen in liquid N$_2$ before being homogenized in ice-cold lysis buffer containing protease and phosphatase inhibitors[48]. Data are shown from $N = 8$ subjects for endurance exercise and $N = 7$ subjects for resistance exercise due to insufficient sample available from one subject. The study was approved by the Human Research Ethics Committee of the RMIT University (Melbourne, Australia) and the Massey University (Palmerston North, New Zealand). Written informed consent was obtained from each participant.

**Western blotting.** Skeletal muscles from mice were rapidly dissected, frozen in liquid N$_2$ and homogenized using a TissueLyser (Qiagen) in a modified RIPA

buffer containing: 50 mM Tris-HCl (pH 7.5), 1 mM EDTA, 10% (v/v) glycerol, 1% (v/v) Triton-X, 0.5% sodium deoxycholate, 0.1% SDS, 1 mM DTT, and protease/phosphatase inhibitor cocktail (Pierce). The protein content of muscle and tissue lysates were analyzed using a Bradford Assay (Biorad), and samples containing equal amounts of protein were heated to 95 °C for 5 min in laemmli buffer. Samples were run on Criterion TGX 4–15% gradient gels (Bio-Rad) and transferred to Nitrocellulose membranes. Stain-free technology (Bio-Rad) was used to determine equal loading. Membranes were blocked in 5% non-fat dry milk or bovine serum albumin (BSA) for 1 h at room temperature and exposed to primary antibodies overnight at 4 C, followed by incubation with appropriate HRP-conjugated secondary antibodies and visualization with ECL on film or a ChemiDoc Touch imaging system (Bio-Rad). The following antibodies were used for the detection of phosphorylated and total protein levels: pSMAD2-C (CST 8828), pSMAD2-L (CST 3104), pSMAD3-L (PA5-38521), pERK (CST 4370), SMAD 2 (CST 5339), SMAD 2/3 (CST 8685), pJNK (Promega V7931 or CST4668), JNK Total (CST 9252), α-Tubulin (CST 3873), GAPDH (CST5174), pAkt (CST 9271), pP38 (CST 4511), pAMPK (CST 2531). All primary antibodies were used at 1:1000 dilution in TBST, with the exception of pSMAD2-L, which was diluted at 1:500. Uncropped images of the most important blots are included in Supplementary Figure 2.

**Real time PCR.** A portion of the Tibialis Anterior muscle was collected from muscle-specific JNK knockout mice and controls and stored for 24 h in RNALater (Qiagen) at 4 °C. RNA was extracted using TRIzol reagent and purified using Direct-zol columns (Zymo). RNA samples underwent reverse transcription and cDNA levels were measured using target-specific primers. Primer sequences used are outlined in Table 1.

**In situ contraction and acute treadmill running.** In situ muscle contraction was performed in ICR mice (Fig. 3) or mJNKKO and control mice (Fig. 4). For these experiments, the sciatic nerve was isolated from one leg and stimulated for 15 min (1 train s$^{-1}$, 500 ms train duration), with the contralateral leg acting as a basal control. Following completion of the contraction protocol, the tibialis anterior muscles from both legs were rapidly dissected and frozen in liquid N$_2$, and subsequently processed for Western Blotting. For acute treadmill running experiments, mice were placed on a motorized treadmill at a speed of 15 m/min (15% grade) for 15, 30, or 60 min. Immediately following exercise, mice were killed by cervical dislocation and the tibialis anterior muscles were rapidly dissected and frozen in liquid N$_2$. Mice that did not undergo treadmill running served as controls.

**Muscle stretch.** Mice were killed by cervical dislocation, the soleus muscles from both hindlimbs were rapidly removed, and the tendon ends were tied with sutures. Muscles were then mounted to a force transducer at resting tension and pre-incubated in oxygenated Krebs-Ringer Bicarbonate Buffer with 2 mM pyruvate for 30 min. Following the pre-incubation period, one muscle from each mouse was stretched at 0.12 N for 10 min, while the contralateral muscle remained at resting tension and acted as a control. Following the stretch experiment, muscles were immediately frozen in liquid N$_2$, and stored and −80 °C until processing for western blotting.

**Tissue culture.** C2C12 myoblasts (ATCC; passage 5–9) or HEK293 cells were used for all experiments and maintained in DMEM containing 10% FBS and 1% Penicillin/Streptomycin. C2C12 myoblasts underwent serum starvation for 2 h, followed by treatment with 5 μM anisomycin for 30 min to determine the effect of JNK activation on SMAD2-linker phosphorylation. To determine whether JNK-mediated SMAD linker phosphorylation is isoform-specific, cells were transfected with plasmids encoding the JNK activating kinase MKK7 fused to either JNK1 or JNK2 isoforms (Addgene plasmids #19726 for JNK1 and #19272 for JNK2;[50]). This fusion plasmid strategy leads to expression of a constitutively active form of JNK[50], referred to as active JNK1 (aJNK1) and active JNK2 (aJNK2). Control cells were transfected with pCDNA3 empty vector. All myoblast transfections were performed using Viromer Red Transfection Reagent (Lipocalyx). For measurement of SMAD transcriptional activity HEK293 cells (ATCC; passage 5–10) were reverse-

transfected with a SMAD-binding element luciferase reporter plasmid (SBE4-Luc; Addgene plasmid #16495 from Bert Vogelstein[51]) in the presence or absence of active JNK1. HEK293 transfections were performed using Lipofectamine 3000 transfection reagent (Thermo Fisher). Luciferase activity was measured using a Pierce Firefly Luciferase Glow Assay kit. Isolation of nuclear SMAD2 was performed using NE-PER Nuclear and Cytoplasmic extraction reagents. In experiments involving myostatin treatment, recombinant myostatin (R&D Systems, Cat#788-G8) was added to cell culture medium at a final concentration of 20 nM. Data are representative of three independent experiments.

**In vitro JNK immunoprecipitation and activity assay.** Fusion plasmids encoding for active, FLAG-tagged, JNK1 and JNK2 were expressed in C2C12 myoblasts for 24 h as described in Tissue Culture. Cells expressing an empty plasmid vector were used as a control. Protein was extracted in a Triton-X lysis buffer (pH 7.5) containing protease and phosphatase inhibitors. Lysates were incubated overnight at 4 °C with anti-FLAG M2 affinity gel (Sigma) to immunoprecipitate active JNK proteins. Immunoprecipitates were washed thoroughly to remove unbound proteins, and incubated for 1 h at 30 °C in a kinase activity buffer containing ATP, and either recombinant human SMAD2 or SMAD3 substrates (Abcam). The reaction was stopped by adding 4X Laemmli buffer and heating at 95 °C for 5 min. The degree of SMAD phosphorylation was determined in the supernatants by western blotting, as described above.

**Muscle histology.** Skeletal muscles from mJNKKO mice and controls were frozen in isopentane cooled with liquid $N_2$ and stored at −80 °C until sectioning. A total of 6 μm sections were cut using a cryostat microtome (Leica CM1850; Leica Microsystems) and affixed to slides. Sections were incubated overnight at 4 °C in a humidified chamber with antibodies against myosin heavy chain type I (A4.951, Developmental Studies Hybridoma Bank, University of Iowa). Subsequently, sections were incubated with appropriate conjugated secondary antibodies for one hour at 37 °C (Alexa Fluor-568, Molecular Probes, Life Technologies). Fluorescent-conjugated (AlexaFluor488) Wheat Germ Agglutinin (WGA) was used to stain the extracellular matrix for the calculation of fiber cross-sectional area using Image J software.

**Statistical analysis.** Analysis of genotype/exercise interactions in animals (Figs. 1 and 2), and JNK/myostatin interactions in cell culture experiments (Fig. 5) were performed using a two-way ANOVA with Sidak's post hoc testing. Analysis of differences in gene expression between wild-type and mJNKKO mice was performed using an unpaired $t$-test (Fig. 4). Human exercise time-course samples were analyzed using a Two-Way ANOVA (Exercise vs. Time) followed by Tukey's multiple comparison test. In all analyses, significance was accepted with $P < 0.05$.

**Data availability.** Data supporting the findings of this study are available within the article and its Supplementary Information files. Additional data may be obtained from the corresponding author upon reasonable request.

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

## Acknowledgements

This work was supported by an American Heart Association grant to S.J.L. (Award #15SDG25560057), and the Joslin Diabetes Center DRC (P30 DK36836). Additional funding was provided by grants to L.J.G. (R01 DK101043) and R.J.D. (R01 DK107220 and R01 DK112698). Work in the human exercise model was funded by the Emerging Researcher Grant Scheme at RMIT University to V.G.C. D.A.R. is supported by grant K01 AG047247. This work was also supported by the U.S. Department of Agriculture (USDA), under agreement No. 58-1950-4-003. R.J.D. is an Investigator of the Howard Hughes Medical Institute.

## Author contributions

Conceptualization, S.J.L., V.G.C., J.E., R.J.D., and L.J.G.; Methodology, S.J.L., M.S.H., M.F.H., and R.J.D.; Investigation, S.J.L., T.L.M., P.P., M.S.H., V.G.C., J.E., D.A.R., and M.F.H.; Formal analysis, S.J.L., T.L.M., and P.P.; Writing, S.J.L., V.G.C., R.J.D., and L.J.G.; Funding acquisition, S.J.L., V.G.C., R.J.D., and L.J.G.

## Additional information

**Competing interests:** The authors declare no competing interests.

