## [Peer Review File · Nature Communications]

Reviewers' comments:

Reviewer #1 (Remarks to the Author):

Authors reported that JNK/Smad2 pathway represented as a novel regulating system for muscle remodeling. They showed that resistance exercise promoted JNK1/2 activation, and subsequent induction of phosphorylation of Smad2 linker domain, in turn, induced Smad2 nuclear localization and Myostatin inhibition leading to increased myofiber size. This system can be seen in human muscle. They concluded that JNK is a molecular switch for muscle remodeling. This paper is potentially interesting, but several concerns should be addressed.

1, Mechanisms underlying JNK activation by resistance exercise should be clarified.

2, Previously, p38 was demonstrated activated by exercise, thus, roles of p38 on muscle mass and Smad2 linker domain phosphorylation should be demonstrated like JNK In Figures 3, 4, 5 and 7. ERK is likely not involved in inducing SMAD2 inhibition, but this should be examined. Then, authors should discuss the similarities or differences of signals among MAPKs in muscle.

3, Authors should demonstrate whether the direct interaction between JNK and Smad2 is required for Smad2 linker domain phosphorylation.

4, Authors previously demonstrated that SMAD3 linker domain was not phosphorylated by exercise. Authors should demonstrate that SMAD2 is the sole factor and SMAD3 is not required for regulating muscle remodeling by using muscle specific SMAD2 knockout. If SMAD3 is also required for muscle remodeling, this should be shown experimentally.

Reviewer #2 (Remarks to the Author):

General Comments

This is an excellent study that will contribute greatly to the literature. The authors have used several well-designed experiments (in animals and humans) to determine intracellular signaling involving JNK, SMAD, and myostatin. My comments are only minor.

Specific Comments

1. In the Introduction and Discussion, I suggest the authors comment at least a little more on the potential of these findings. For example, there have been several studies showing an incompatibility between high-intensity endurance and resistance training. One theory postulates competing adaptations of intracellular signaling. The authors' results help piece together this potential incompatibility and could be used, in part, as a proposed mechanism of competition. The authors do briefly mention the incompatibility in the Results section but I believe further discussion is warranted.

2. The "Human Subjects" section could use additional information. Albeit this is not an exercise science journal per se, I feel it is important to include more information on the training status of the subjects and the protocols. The authors report subject characteristics were reported elsewhere, but were these trained subjects or untrained? If trained, how much RE or aerobic exercise experience in years? A few lines of the VO₂ peak testing and 1RM testing would be helpful, how long (in days) before the protocols were subjects tested? For the leg extension exercise, how much rest was allowed in between sets? Were the biopsies obtained from one leg or both since there were 5?

We thank the reviewers for their insightful comments and suggestions regarding our manuscript. Significant changes have been made to the manuscript text, and several new experiments have been performed to address the reviewer concerns. We believe the changes made have significantly improved the manuscript. A detailed response to each comment is provided below and changes within the manuscript text are highlighted with red font.

Reviewer #1:

1. *“Mechanism underlying JNK activation by resistance exercise should be clarified.”*

Muscle contractions may activate JNK via either mechanical stress or metabolic stress. While endurance exercise induces primarily metabolic stress in muscle, resistance exercise induces significant mechanical stress, which is necessary to stimulate muscle hypertrophy. Given that JNK is more robustly activated by resistance exercise in humans, we hypothesize that the additional mechanical strain of resistance exercise (vs. endurance exercise) is the mechanism underlying JNK activation by this modality. To test this hypothesis, we performed new experiments to determine the level of JNK activation, along with the activation of several other stress-activated kinases in muscle in response to mechanical stress. Passive muscle stretch induces mechanical stress, but relatively little metabolic stress in muscle. Our results demonstrate that passive stress can robustly activate JNK and SMAD2-L phosphorylation, but does not consistently activate the metabolic stress indicators, AMPK and ERK (Figure 3C). This finding is consistent with our previous data showing that JNK is only mildly activated by endurance exercise in mice, but was strongly activated by in situ contraction, which elicits significant mechanical stress via eccentric contraction. We have now added data demonstrating that JNK/SMAD signaling is activated by passive stretch to Figure 3. In addition, we have modified the results and discussion of the manuscript to reflect our hypothesis that JNK/SMAD signaling is likely activated by resistance exercise due to increased mechanical stress. Changes to the text are found in the Results Section, Page 5; the Discussion, Page 9; and the Methods, Page 12-13.

2. *“Previously, p38 was demonstrated to be activated by exercise, thus roles of p38 should be demonstrated like JNK...ERK is likely not involved in inducing SMAD2 inhibition, but this should be examined...authors should discuss the similarities or differences of signals among MAPKs in muscle.”*

We have performed additional Western Blotting experiments to include the activation of p38 and ERK with various modes of muscle stimulation/contraction in Figure 3, and in C2C12 experiments with Anisomycin and active JNK expression (Figure 4D & E). In muscle, we found that JNK activation tracked with pSMAD2L phosphorylation with treadmill running, in situ contraction, and muscle stretch. In contrast, we found that p38 MAPK was activated with in situ contraction and stretch, but not treadmill running. ERK appeared to be activated in some treadmill running samples, but did not track with pSMAD2L phosphorylation. Unlike P38 and JNK, ERK did not seem to be responsive to mechanical stress, since it was minimally activated by passive muscle stretch. A similar pattern to ERK activation was found with the metabolic sensor, AMPK, which was also not activated by muscle stretch. Thus, these additional experiments indicate that only JNK is activated in parallel with pSMAD2L phosphorylation in all

3 models, supporting our hypothesis that JNK is the primary upstream kinase for pSMAD2L in muscle. These new data have been added to Figure 3, and the results section on Page 5.

In additional experiments performed in C2C12 myoblasts, we found that Anisomycin treatment activated JNK, p38 and ERK to the same extent, since Anisomycin is not a specific JNK activator (Figure 4D). To determine whether JNK is sufficient to activate SMAD2L phosphorylation, we expressed active JNK1 and/or JNK2 in C2C12 cells. We found that SMAD2L phosphorylation was increased by expression of both JNK1 and JNK2, but this was not associated with increased activation of p38 or ERK (Figure 4E). In addition, we demonstrated that active JNK purified by immunoprecipitation was able to phosphorylate SMAD2 in the linker region, in the absence of any other cellular proteins. Our previous results also showed that SMAD2L phosphorylation with contraction was largely absent in muscle-specific JNK1/2 knockout mice (Figure 4A-C). Thus, we demonstrate that JNK is both necessary and sufficient for SMAD2L phosphorylation in muscle. When considered collectively, our data provides strong evidence that JNK is the primary upstream kinase responsible for SMAD2L phosphorylation with muscle contraction. These new results have been added to Figure 4 and the Results, Page 6.

3. *Authors should demonstrate whether the direct interaction between JNK and SMAD2 is required for SMAD2 linker domain phosphorylation*

To address this concern, we have performed a JNK activity assay using immunoprecipitates of lysates expressing active JNK incubated with recombinant SMAD2 protein (Figure 4F). We found that incubating recombinant SMAD2 protein with purified anti-FLAG immunoprecipitates for either JNK1 or JNK2 significantly increased SMAD2L phosphorylation, compared to control immunoprecipitates from lysates expressing an Empty Vector (PCDNA3.1). These experiments indicate that JNK can directly phosphorylate SMAD2 in vitro, in the absence of any other cellular proteins. In addition, we demonstrate that this effect is specific to SMAD2, as no phosphorylation was noted in reactions containing recombinant SMAD3 (Figure 4F). This additional data has been added to Figure 4, as well as the Results section on Page 6. Methods describing the in vitro JNK activity assay have been added to Page 13.

4. *Authors previously demonstrated that SMAD3 linker domain was not phosphorylated by exercise. Authors should demonstrate that SMAD2 is the sole factor and SMAD3 is not required for regulating muscle remodeling by using muscle specific SMAD2 knockout. If SMAD3 is also required for remodeling, this should be shown experimentally.*

Work by other laboratories has demonstrated that both SMAD2 and SMAD3 are indeed required for the maintenance of muscle mass, and TGF β /Myostatin activity (Sartori et al., 2009; Trendelenburg et al., 2009; Tando et al., 2016). These investigations have demonstrated that SMAD2 and SMAD3 play a redundant role in the regulation of muscle mass, as knockdown of either isoform alone is insufficient to change muscle phenotype, while knockdown of both isoforms simultaneously has a positive effect on muscle mass. We have added the above citations to the manuscript to describe the essential role of SMAD3 in Myostatin activity (Discussion, Page 9).

To determine whether JNK may also phosphorylate SMAD3 in its linker region, we performed Western blotting analysis of the skeletal muscle samples used in Figures 3 and 4 using a commercially available antibody against the SMAD3 linker region (pSMAD3 Ser208; ThermoFisher PA5-38521). Although other phosphorylation sites have been identified in the SMAD3 linker region (Thr179, Ser203, Ser213), the Ser208 site is predicted to be the primary MAPK target site. In all of our experiments, we were unable to detect SMAD3 phosphorylation at the Ser208 site with contraction, exercise, or JNK activation. As noted by the reviewer, this result is consistent with our previous results in rat muscle (Lessard et al., Diabetes, 2013). While a potentially non-specific band appeared at ~30kDa using the pSMAD3 Ser208 antibody, no detectable band was seen at the predicted and observed molecular weight of SMAD3 (~52 kDa), even if long (i.e. 20 minute) exposures were used. An example of blots using the pSMAD2L and pSMAD3L antibodies on muscle basal (B) and stretched (S) muscle samples from Figure 3C is shown below:

In addition, we performed an *in vitro* experiment to determine whether purified JNK can directly phosphorylate recombinant SMAD2 and SMAD3. While we observed an increase in SMAD2L phosphorylation following incubation with either JNK1 or JNK2 in these experiments, no detectable phosphorylation of SMAD3 in the linker region was detected using the pSMAD3 Ser208 antibody. The catalogue number for the antibody that was used to attempt detection of SMAD3 linker phosphorylation has been added to the methods (Page 12).

There are several possible reasons for the absence of SMAD3 linker phosphorylation in these experiments:

- i) JNK may phosphorylate SMAD3 at an alternative site to Ser208 either within, or outside of the linker region with muscle contraction which has not yet been discovered.

- ii) Antibodies generated for the SMAD2 linker region (CST #3104) are more sensitive than currently available antibodies generated for the SMAD3 linker region. The pSMAD2L that we are using can detect phosphorylation at 3 consecutive linker region residues (Ser245/250/255). Unfortunately, no similar antibody for SMAD3 is currently available.
- iii) JNK does not phosphorylate SMAD3 in skeletal muscle with exercise.

Although we are unable to observe specific changes in SMAD3 linker phosphorylation with any of our muscle contraction or C2C12 models, we did observe that active JNK overexpression caused an impairment of both SMAD2 and SMAD3 translocation to the nucleus in C2C12 (Figure 5C). These results suggest that SMAD3 likely contributes to the reduction in Myostatin transcription with increased JNK activity. We have now highlighted this result in the Results (Page 7) and Discussion (Page 8-9) to indicate that we provide experimental evidence that JNK inhibits both SMAD2 and SMAD3 activity. We also propose that JNK may affect SMAD3 activity indirectly via phosphorylation of SMAD2, since these two isoforms can form hetero-trimers along with SMAD4 and translocate the nucleus together (Lucarelli et al., 2018). While we have not observed evidence of JNK-mediated SMAD3 phosphorylation in our current and previous investigations, we are fully confident in our identification of SMAD2 as a specific target of JNK with exercise, and the ability of this phosphorylation event to act as a marker of reduced Myostatin activity. We feel the publication of this novel exercise-activated signaling cascade will be of great interest to other researchers in the field, and that future research will further elucidate the specific targets and functions of this important pathway.

Reviewer #2:

1. *“In the Introduction and Discussion, I suggest the authors comment a little more on the potential of these findings...the authors do briefly mention incompatibility in the Results, but I believe further discussion is warranted.”*

We have expanded the Introduction and Discussion of the manuscript to address the significance of our data with respect to the incompatibility of concurrent endurance and resistance training. Changes appear in the first paragraph of the Introduction (Page 2), and the second paragraph of the Discussion (Page 8). Original text describing concurrent training interference appears in the Results (Page 5).

2. *“The Human Subjects section could use additional information...on the training status of the subjects and the protocols...a few lines of VO_{2peak} and 1RM testing would be helpful...were biopsies from one leg or both?...”*

To address this concern, we modified the Human Subjects Methods to include more detail regarding the available training history of the subjects (recreationally active in various sports), and the specific protocols used for VO_{2peak} and strength testing. We have also included average VO_{2peak} and 1RM values for each experimental group to provide readers with an idea of the subject fitness levels. In addition, we have clarified that the biopsies were taken from separate incisions (distal to proximal), from the same leg. This additional information has been added to Page 11-12 of the Methods section.

REVIEWERS' COMMENTS:

Reviewer #1 (Remarks to the Author):

Authors added new data, and the manuscript was improved. However, two points raised by this reviewer at the initial submission remained to be addressed as below.

1, Mechanisms underlying JNK activation by resistance exercise are not understandable. Authors should demonstrate how JNK senses the exercise stimuli for phosphorylation. Are some molecules required to transfer the exercise signals to activate JNK?

2, p38 and ERK data should be shown in Figure 5 and 7.

Reviewer #2 (Remarks to the Author):

The authors have addressed my concerns. I have no further comments.

RESPONSES TO REVIEWER COMMENTS: Author responses are in **bold**.

REVIEWERS' COMMENTS:

Reviewer #1 (Remarks to the Author):

Authors added new data, and the manuscript was improved. However, two points raised by this reviewer at the initial submission remained to be addressed as below.

1, Mechanisms underlying JNK activation by resistance exercise are not understandable. Authors should demonstrate how JNK senses the exercise stimuli for phosphorylation. Are some molecules required to transfer the exercise signals to activate JNK?

As suggested by the editor, we have not addressed comment 1.

2, p38 and ERK data should be shown in Figure 5 and 7.

The purpose behind the reviewer's request for us to show p38 and pERK in the previous revision was to provide evidence that JNK, and not another MAP-kinase, is the primary kinase responsible for pSMAD2L phosphorylation. To address this, we performed p38 and pERK blots for all contraction exercise models in Figure 3, and all tissue culture models in Figure 4. This additional work demonstrated that only JNK phosphorylation (but not pP38 and pERK) correlated with pSMAD2L phosphorylation in vivo with exercise and contraction. We also demonstrated that active JNK overexpression in myoblasts (which is a model also used in Figure 5) does not result in P38 and ERK activation, and therefore effects shown with this model are JNK-specific. Finally, our results from muscle-specific JNK knockout mice clearly demonstrate that JNK is the primary upstream kinase for SMAD2L phosphorylation, as SMAD2L phosphorylation with contraction is largely abolished in JNK knockout mice. Although the reviewer requests addition of pP38 and pERK to Figure 5, we feel this would be redundant, as we have already demonstrated that the active JNK overexpression model does not lead to P38 and ERK activation in Figure 4. The primary purpose of Figure 5 is to show the changes in Myostatin activity and SMAD translocation with JNK activation. We feel addition of p38 and pERK data to this already complex figure would detract from our main findings. Therefore, we have not added p38 and pERK blots to Figure 5. For Figure 7, we only had sufficient sample to run one additional blot on our human samples. Therefore, we did add pP38 to Figure 7, since like JNK, P38 was found to be activated by mechanical stress in muscle (Figure 3C). We were unable to obtain clear results for pERK using the same membrane as P38 because their molecular weights run very close to each other (38 vs. 42/44 kDa). However, as mentioned previously, our work in animal and tissue culture models clearly show that JNK is the upstream kinase for pSMAD2L phosphorylation. Our data from Figure 7 supports this assertion in humans, as pJNK and pSMAD2L activation were highly correlated in skeletal muscle from human subjects (Figure 7E).